# Computing all Optimal Partial Transports

**Abhijeet Phatak[1], Sharath Raghvendra[2], Chittaranjan Tripathy[1], Kaiyi Zhang[2]**[*]
[1]Walmart Global Tech, [2]Virginia Tech
{abhijeet.phatak,ctripathy}@walmart.com
{sharathr,kaiyiz}@vt.edu

## Abstract

We consider the classical version of the optimal partial transport problem. Let $\mu$ (with a mass of $U$) and $\nu$ (with a mass of $S$) be two discrete mass distributions with $S \leq U$ and let $n$ be the total number of points in the supports of $\mu$ and $\nu$. For a parameter $\alpha \in [0, S]$, consider the minimum-cost transport plan $\sigma_\alpha$ that transports a mass of $\alpha$ from $\nu$ to $\mu$. An *OT-profile* captures the behavior of the cost of $\sigma_\alpha$ as $\alpha$ varies from $0$ to $S$. There is only limited work on OT-profile and its mathematical properties (see Figalli (2010)). In this paper, we present a novel framework to analyze the properties of the OT-profile and also present an algorithm to compute it. When $\mu$ and $\nu$ are discrete mass distributions, we show that the OT-profile is a piecewise-linear non-decreasing convex function. Let $K$ be the combinatorial complexity of this function, i.e., the number of line segments required to represent the OT-profile. Our exact algorithm computes the OT-profile in $\tilde{O}(n^2 K)$ time. Given $\delta > 0$, we also show that the algorithm by Lahn et al. (2019) can be used to $\delta$-approximate the OT-profile in $O(n^2/\delta + n/\delta^2)$ time. This approximation is a piecewise-linear function of a combinatorial complexity of $O(1/\delta)$. An OT-profile is arguably more valuable than the OT-cost itself and can be used within applications. Under a reasonable assumption of outliers, we also show that the first derivative of the OT-profile sees a noticeable rise before any of the mass from outliers is transported. By using this property, we get an improved prediction accuracy for an outlier detection experiment. We also use this property to predict labels and estimate the class priors within PU-Learning experiments. Both these experiments are conducted on real datasets.

## 1 Introduction

Given two discrete probability distributions $\mu$ (with a mass of $U = 1$) with the set $A$ as the support and $\nu$ (with a mass of $S = 1$) with $B$ as the support, where $|A| + |B| = n$, in the *optimal transport* problem, one wishes to compute the minimum cost plan to transport mass from $\nu$ to $\mu$. When the mass $U \neq S$, the problem is called the *unbalanced optimal transport*. In the *partial optimal transport* problem, given a parameter $\alpha \in [0, S]$, one wishes to determine the $\alpha$-optimal partial transport cost which is the minimum work required to transport a mass of $\alpha$ from $\nu$ to $\mu$.

Owing to its strong statistical properties, the optimal transport cost (Villani (2003); Peyré & Cuturi (2019)) is considered to be an attractive dissimilarity metric between probability distributions, and has found numerous applications in areas involving GANs, image processing, (Arjovsky et al. (2017); Liu et al. (2018); Balaji et al. (2020); Lin et al. (2021); Schmitz et al. (2018); Chen et al. (2019)), variational inference (Ambrogioni et al. (2018)), econometrics (Galichon (2016)) and other areas of natural science (Schiebinger et al. (2019); Sun et al. (2020)) and applied mathematics (Santambrogio (2015)). Similarly the unbalanced and partial optimal transport has been used for various problems that arise in machine learning, including GAN training, image processing, outlier detection and Positive Unlabelled (PU-) learning (Yang & Uhler (2018); Bonneel & Coeurjolly (2019); Chapel et al. (2020); Mukherjee et al. (2021)).

The exact optimal transport cost and plan, including in the unbalanced case, can be computed in $O(n^3 \log n)$ time. For a fixed value of $\alpha \in [0, S]$, one can easily reduce the problem of computing an

---

[*]Following convention from Theoretical Computer Science, all authors are ordered in alphabetical order

$\alpha$-optimal partial transport cost to solving an unbalanced instance of the optimal transport problem in $O(n^3 \log n)$ time: Create a *catchment* node $r$ in the support of $\mu$ with a additional mass of $(S - \alpha)$. Let $A' = A \cup \{r\}$ be the new support of $\mu$. Note that the total mass of $A'$ is $U + S - \alpha$. For every $b \in B$, add an edge $(r, b)$ with a cost of $0$. To find the $\alpha$-optimal partial transport, one can simply solve the unbalanced optimal transport between $A'$ and $B$ using an exact solver in $O(n^3 \log n)$ time.

Several algorithms approximate the optimal transport cost in $O(n^2 \text{poly}\{1/\delta, \log n\})$ time. Cuturi (2013) introduced the Sinkhorn algorithm to solve the entropic regularized optimal transport problem and showed that it can be used to approximate the optimal transport cost within an additive factor of $\delta$ in $\tilde{O}(n^2/\delta^2)$ time; see also Abid & Gower (2018); Altschuler et al. (2017); Dvurechensky et al. (2018); Lin et al. (2019); Guo et al. (2020); Xie et al. (2022). Since then, there has been significant research on the design of additive approximation algorithms and several algorithms achieve an execution time of $\tilde{O}(n^2/\delta)$ (Lahn et al. (2019); Jambulapati et al. (2019); Quanrud (2019)). The state-of-the-art execution time for approximating the Optimal Transport is achieved by the combinatorial algorithm by Lahn et al. (2019) (LMR-algorithm). Their algorithm is based on adapting a classical graph theory algorithm Gabow & Tarjan (1989) and runs in $O(n^2/\delta + n/\delta^2)$ time. In this paper, we study the classical version of the optimal partial transport problem which we introduce next.

We are given $\mu$ and $\nu$ whose supports are the point sets $A$ and $B$, respectively. Let $G(A \cup B, A \times B)$ be a complete bipartite graph with $A \cup B$ as the vertex set and $A \times B$ as its set of edges. For any $a \in A$ (resp. $b \in B$), we associate a mass of $\mu_a$ (resp. $\nu_b$) such that the $U = \sum_{a \in A} \mu_a$ and $S = \sum_{b \in B} \nu_b \leq U$ [1]. We refer to each point $a \in A$ (resp. $b \in B$) to be a demand (resp. supply) point and assume $\mu_a$ (resp. $\nu_b$) to be a positive rational number. For any pair of points $a \in A$ and $b \in B$, we are given a non-negative cost $c(a, b) \in \mathbb{R}_{\geq 0}$ bounded by $1$. The cost of transporting a supply of mass $\beta$ from $b$ to $a$ is $\beta c(a, b)$. A transport plan is a function $\sigma : A \times B \to \mathbb{R}_{\geq 0}$ that assigns a non-negative value to each edge of $G$ indicating the quantity of supply transported along the edge. The transport plan $\sigma$ is such that the total supplies transported into (resp. from) any demand (resp. supply) node $a \in A$ (resp. $b \in B$) is bounded by the demand $\mu_a$ (resp. supply $\nu_b$) at $a$ (resp. $b$), i.e., $\sum_{b \in B} \sigma(a, b) = \mu_a$ (resp. $\sum_{a \in A} \sigma(a, b) = \nu_b$). For any $\alpha \in [0, S]$, we say that any transport plan $\sigma$ is an $\alpha$-partial transport plan if it transports a mass of $\alpha$ from $\nu$ to $\mu$, i.e., $\sum_{(a,b) \in A \times B} \sigma(a, b) = \alpha$. The cost of the partial transport plan denoted by $w(\sigma)$ is given by $\sum_{(a,b) \in A \times B} \sigma(a, b) c(a, b)$. In the $\alpha$-optimal partial transport problem, we are interested in finding a minimum-cost $\alpha$-partial transport plan which we denote by $\sigma_\alpha^*$. We define the *OT-profile* to be a function $\omega : [0, S] \to \mathbb{R}_{\geq 0}$ that maps a value $\alpha \in [0, S]$ to the cost of $w(\sigma_\alpha^*)$ as $\alpha$ goes from $0$ to $S$. For discrete distributions, we show that this function is convex and piecewise-linear. Let $K$ denote the combinatorial complexity of $\omega$. Since, OT profile is a piecewise-linear convex function, its first derivative is a non-decreasing step-function. We denote this step function as $D\omega$ where $D\omega(\alpha)$ denotes the first derivative of the OT-profile at $\alpha$.

Next, we define the notion of a $\delta$-*approximate OT-profile*. We say that a function $\overline{\omega} : [0, S] \to \mathbb{R}_{\geq 0}$ $\delta$-approximates the OT-profile, if, for every $\alpha \in [0, S]$, $\omega(\alpha) \leq \overline{\omega}(\alpha) \leq \omega(\alpha) + S\delta$. Recollect that, when $\mu$ and $\nu$ are probability distributions, $U = S = 1$ and we get $\omega(\alpha) \leq \overline{\omega}(\alpha) \leq \omega(\alpha) + \delta$. Therefore, $\overline{\omega}(\alpha)$ is an additive approximation of $\omega(\alpha)$ and the function $\overline{\omega}$ represents an additive approximations of all optimal partial transports.

From a theoretical standpoint, there is limited understanding of the properties of an OT-profile and its first derivative. In the seminal work by Figalli (2010), he considered the OT-profile of the optimal partial transports for the case where the distributions are continuous and the ground distance $c(u, v)$ is $\|u - v\|^2$. To the best of our knowledge, we are not aware of any other work on OT-profile.

**Our Contributions:** In this paper, we present a new exact and an approximation algorithm to compute the OT-profile. We also provide a novel framework to derive properties of the OT-profile and its first derivative. Using this framework, we show how an OT-profile can be used to identify points from the outlier class and the inlier class within the support of a distribution. All our results apply for any arbitrary cost function $c(\cdot, \cdot)$.

- First, we present a simple primal-dual based combinatorial algorithm to compute the exact optimal transport cost. Our algorithm is a generalization of the well-known Hungarian method for the

---

[1] We consider the generalized case for arbitrary and potentially unbalanced case. For the case where $\mu$ and $\nu$ are probability distributions $U = 1$ and $S = 1$

assignment problem. Interestingly, we show that this algorithm not only computes the OT-cost but also traces the entire OT-profile (Lemma 2.1 and 3.1). Moreover, as the OT-profile is traced, certain dual weights maintained by the algorithm capture the first derivative values (Lemma 3.4). By tracking the evolution of these dual weights, we show that the first derivative $D\omega$ is a non-decreasing step function (Lemma 3.4 and 3.2(b)) and therefore, $\omega$ is piecewise-linear convex function. Our algorithm constructs $\omega$ and $D\omega$ in $\tilde{O}(n^2 K)$ time; here $K$ is the *combinatorial complexity* of $\omega$, i.e., the number of line segments in $\omega$.

- We further exploit these properties (Lemma 3.4 and 3.2) to show that the OT-profile can be used to detect outliers. Recollect $A$ and $B$ are supports of the distributions $\mu$ and $\nu$, respectively. ]Let $A$ consists of points from the 'inlier' class (with a total mass of $U$) and $B$ be a dataset that contains both points $B^+$ from the inlier class (with a total mass of $\alpha^*$) and points $B^-$ from the outlier class (with a total mass of $S - \alpha^*$). Let $w$ denote the minimum-cost required to transport mass from $B^+$ to $A$. We assume that outlier points in $B^-$ are separated from the inlier points of $A$ as follows: *Assumption:* For some small $\varepsilon > 0$ and a constant $C > 1$, every outlier point $b \in B^-$ is at a distance at least $Cw/\varepsilon$ from any inlier point in $A$. Under this assumption of the outlier points, we show the following lemma

  **Lemma 1.1** (Outlier Lemma). *(A) The $(\alpha^* - \varepsilon)$-optimal partial transport generated by our algorithm will not transport any mass from the outlier points, and, (B) The first derivative values $D\omega(\alpha^* - \varepsilon) \leq w/\varepsilon$ and $D\omega(\alpha^* + \varepsilon) \geq Cw/\varepsilon$.*

  In other words, the first derivative function $D\omega$ will show a noticeable rise in an interval of width $\varepsilon$ around $\alpha^*$. By detecting this rise, we can approximate the values $\alpha^*$ and also mark the inliers and outliers. The proof of Lemma 1.1 is presented in the Appendix (See Section 7.5).

- We prove that the current state-of-the-art algorithm (LMR-algorithm) to approximate the OT cost can also generate a $\delta$-approximation $\overline{\omega}$ of the OT-profile. The combinatorial complexity of $\overline{\omega}$ is only $\lceil 4/\delta \rceil$. Thus, one can not only approximate the OT-cost but construct an approximate OT-profile in $O(n^2/\delta + n/\delta^2)$ time. However, $\overline{\omega}$ is not a convex function, its first derivative may be very unstable and therefore does not satisfy any approximate version of the Outlier Lemma (Lemma 1.1). Instead, by drawing insights from our exact algorithm, we describe a non-decreasing step function $D\overline{\omega}$ that tracks the evolution of certain dual weights maintained by the LMR-algorithm (see Section 8.2.1). We then show that $D\overline{\omega}$ satisfies an approximate version of the Outlier Lemma (Lemma 8.3).

- We consider two applications where partial optimal transport has been successful and show how an OT-profile can help in improving accuracy while reducing the need for data dependent parameters. First, we conduct an outlier detection experiment that was proposed by Mukherjee et al. (2021). In this, we are given two sets of MNIST images, one set is the inlier (digits 0-4) the other is the mixture containing $\alpha^*$-fraction of inliers (0-4) and $(1 - \alpha^*)$-fraction outlier (5-9) data. We can obtain $\alpha^*$ by identifying a sharp rise in the first derivative function $D\omega$. Then, we mark the free vertices with respect to $\alpha^*$-optimal partial transport as outliers. In contrast, the method proposed in Mukherjee et al. (2021) depends on an additional data driven parameter that approximates the average distance between inliers. We conduct a second set of experiments for PU-learning that was proposed in Chapel et al. (2020). In this experiment, we are given a small set of positive samples and an unlabelled dataset. We are required to use them and identify (a) the fraction $\alpha^*$ of positive samples in the unlabelled set (called the *class prior*), and, (b) classify all the samples in the unlabelled dataset as positive or negative. Chapel et al. (2020) presented a optimal partial transport based method that assumed the knowledge of class prior. In contrast, we use OT-profile to both identify the class prior $\alpha^*$ as well as generate a set of labels for the unlabelled data. In Figure 1, we show the first derivative of the OT-profile for four real-world PU-Learning experiments (See Section 5.2). In each case, one can observe a noticeable jump around the class prior. In both outlier detection as well as PU-Learning experiments, the use of OT-profile leads to a better prediction accuracy as well as a reduced dependence on additional data dependent parameters.

## 2 Preliminaries

In this section, we present notations and definitions that are required to describe the exact algorithm. Recollect $\mu$ and $\nu$ are the distribution of demand and supply, respectively. Consider any transport plan $\sigma$ transporting mass from $\nu$ to $\mu$. We say that a vertex $a \in A$ (resp. $b \in B$) is *free* with respect $\sigma$ if its demand (resp. supply) is not exhausted by $\sigma$, i.e., $\mu_a - \sum_{b \in B} \sigma(a, b) > 0$ (resp.

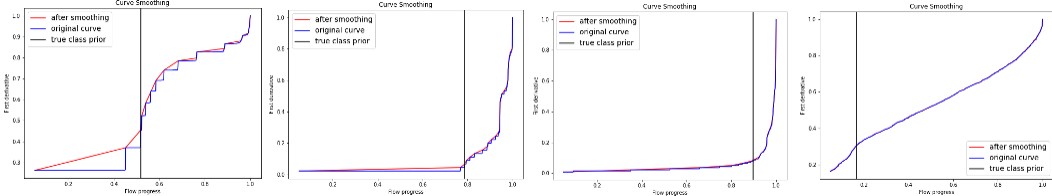

Figure 1: First derivative of the OT-profile for PU-learning Datasets (mushroom, shuttle, pageblocks, and, usps). Notice a jump in the first derivative values near the true class prior $\alpha^*$.

$\nu_b - \sum_{a \in A} \sigma(a, b) > 0$). For free vertices, we refer to $\mu_a - \sum_{b \in B} \sigma(a, b)$ (resp. $\nu_b - \sum_{a \in A} \sigma(a, b)$) as the *deficit* (resp. *surplus*) at $a$ (resp. $b$). At any stage in our algorithm, we use $A_F$ (resp. $B_F$) to denote the set of free demand nodes (resp. free supply nodes). Recollect that $w(\sigma)$ is the cost of any transport plan $\sigma$ with respect to $c(\cdot, \cdot)$.

We say that a transport plan $\sigma$ is *dual feasible* if, for every $a \in A$ and $b \in B$, there are dual weights $y(a)$ and $y(b)$ associated with $a$ and $b$ respectively such that

$$y(a) + y(b) \le c(a, b) \qquad\qquad \text{if } \sigma(a, b) < \min\{\mu_a, \nu_b\} \qquad (1)$$
$$y(a) + y(b) \ge c(a, b) \qquad\qquad \text{if } \sigma(a, b) > 0. \qquad (2)$$

These are the classical constraints corresponding to the dual formulation of the optimal transport problem. Let $y_{\max} = \max_{v \in A \cup B} |y(v)|$. In Lemma 2.1 (see Section 6 for a proof), we show that any feasible transport plan $\sigma$ transporting a mass of $\alpha$ that satisfies the following property is an $\alpha$-optimal partial transport plan:

(C) For every demand (resp. supply) node $a$ (resp. $b$), the dual weight $y(a) \le 0$ (resp. $y(b) \ge 0$) and, if $a$ is a free demand (resp. free supply) node, then $y(a) = 0$ (resp. $y(b) = y_{\max}$).

To compute the OT-profile, therefore, we incrementally construct a transport plan while maintaining (C) as an invariant. By doing so, we guarantee that every intermediate transport plan is also an optimal partial transport plan. We can then use these intermediate transport plans to build the OT-profile.

**Lemma 2.1.** *Let $\sigma$ along with dual weights $y(\cdot)$ be a feasible partial transport plan that transports a mass of $\alpha$ and satisfies (C). Then, $\sigma$ is an $\alpha$-optimal partial transport plan.*

Next, we introduce residual graphs that are extensively used in the design of combinatorial minimum-cost flow algorithms. We use $\overrightarrow{uv}$ to denote an edge directed from a vertex $u$ to a vertex $v$. Given a transport plan $\sigma$, one can build a *residual graph* $\overline{G}_\sigma$ as follows. The residual graph $\overline{G}_\sigma(A \cup B, \overrightarrow{E})$ is a directed graph with $A \cup B$ as the vertex set. For every $a \in A$ and $b \in B$,

- If $\min\{\mu_a, \nu_b\} > \sigma(a, b) \ge 0$, there is a *forward edge* $\overrightarrow{ba}$ to $\overrightarrow{E}$ and assign its residual capacity to be $\min\{\mu_a, \nu_b\} - \sigma(a, b)$.

- If $0 < \sigma(a, b) \le \min\{\mu_a, \nu_b\}$, we add a *backward edge* $\overrightarrow{ab}$ to $\overrightarrow{E}$ with a residual capacity of $\sigma(a, b)$.

We would like to note that if $\sigma(a, b) = 0$ (resp. $\sigma(a, b) = \min\{\mu_a, \nu_b\}$), then there is only a forward (resp. backward) edge between $a$ and $b$ in $\overrightarrow{E}$. In all other cases, both the forward edge $\overrightarrow{ba}$ and the backward edge $\overrightarrow{ab}$ are in $\overrightarrow{E}$. We set the *cost* of any edge between $a$ and $b$ regardless of their direction to be $c(a, b)$. Given a partial transport plan $\sigma$, any directed path in the residual network $\overline{G}_\sigma$ that starts from a free supply vertex $b \in B_F$ and ends at a free demand vertex $a \in A_F$ is called an *augmenting path*. Note that, since the graph is bipartite, edges on any path alternate between a forward edge (directed from $B$ to $A$) and a backward edge (directed from $A$ to $B$). Any augmenting path $P$ in $\overline{G}_\sigma$ can be used to increase the total mass that is transported along $\sigma$. We describe this process of *augmenting* $\sigma$ along the path $P$ next. For any augmenting path $P$ from a free supply vertex $b$ to a free demand vertex $a$, let $s_b$ be the surplus at $b$ and $d_a$ be the deficit at $a$. We denote the bottleneck edge capacity $\kappa$ of an augmenting path $P$ to be the smallest residual capacity among all edges of $P$. Let

$r_P = \min\{\kappa, s_b, d_a\}$ be the *bottleneck capacity* of the path $P$. We can augment $0 < k \le r_P$ units of mass along an augmenting path $P$ as follows. For every forward edge $\overrightarrow{ba}$ on the path $P$, we increase the mass transported from $b$ to $a$ to $\sigma(a,b) \leftarrow \sigma(a,b) + k$. For every backward edge $\overrightarrow{ab}$ on the path $P$, we reduce the mass transported from $b$ to $a$ to $\sigma(a,b) \leftarrow \sigma(a,b) - k$. We show that the updated $\sigma$ continues to be a valid and feasible plan that transports an additional mass of $k$ (See Lemma 2.2). Hence, we also refer to this operation as *pushing* a mass of $k$ along $P$. Finally, we define the *net-cost* of an augmenting path $P$ as

$$\Phi(P) = \sum_{\overrightarrow{uv} \in P \text{ is forward}} c(u,v) - \sum_{\overrightarrow{uv} \in P \text{ is backward}} c(u,v).$$

By its definition, $\Phi(P)$ is simply the change in the cost of the transport plan when we push one unit of mass along $P$. Using this observation, we can show the following (see Section 6 for proof):

**Lemma 2.2.** *For any transport plan $\sigma_\alpha$ that transports a mass of $\alpha$, let $P$ be an augmenting path in the residual graph and let $r_P$ be its residual capacity. For any $k \in [0, r_P]$, let $\sigma_{\alpha+k}$ be the transport plan obtained after pushing $k$ supplies along $P$. Then $\sigma_{\alpha+k}$ is a valid transport plan with a cost $w(\sigma_{\alpha+k}) = w(\sigma_\alpha) + k\Phi(P)$.*

For any $(a,b) \in A \times B$, the *slack* of any forward edge $\overrightarrow{ba}$ in the residual network is

$$s(a,b) = c(a,b) - y(a) - y(b). \tag{3}$$

Any backward edge $\overrightarrow{ab}$ in the residual network has a slack of

$$s(a,b) = y(a) + y(b) - c(a,b). \tag{4}$$

For a feasible transport plan, all slacks are non-negative. Furthermore, note that if both $\overrightarrow{ab}$ and $\overrightarrow{ba}$ are present in the residual network, then the slack is $0$. The next lemma, whose proof is in Section 6, relates the net-cost of any augmenting path to the slack of its edges.

**Lemma 2.3.** *Consider any augmenting path $P$ with respect to a feasible transport plan $\sigma$. Suppose $P$ starts at a free supply vertex $b \in B_F$ and ends at a free demand vertex $a \in A_F$. Then,*

$$\Phi(P) = y(b) - y(a) + \sum_{\overrightarrow{uv} \in P} s(u,v). \tag{5}$$

Next, we define any edge $(a,b)$ in the residual graph $\overline{G}_\sigma$ as *admissible* if $s(a,b) = 0$. The *admissible graph* $\overline{A}_\sigma$ is the subgraph of $\overline{G}_\sigma$ consisting of the admissible edges of the residual graph.

**Lemma 2.4.** *Consider a feasible transport plan $\sigma$ that satisfies (C) and let $P$ be an augmenting path consisting of admissible edges that starts at a surplus node $b \in B$ and ends at a deficit node $a \in A$. Then,*

$$\Phi(P) = y_{\max}. \tag{6}$$

Finally, given a residual network $\overline{G}_\sigma$, we define an *augmented residual network*, $\overline{\mathcal{G}}_\sigma(\mathcal{V}, \mathcal{E})$ as follows. In addition to $A \cup B$, the vertex set $\mathcal{V}$ also contains two additional special vertices $s$ and $t$. The edge set $\mathcal{E}$ consists of two types of edges: (a) Every edge $\overrightarrow{uv} \in \overrightarrow{E}$ is also in $\mathcal{E}$. The *weight* of this edge is set to its slack $s(u,v)$, and (b) For every free supply (resp. demand) node $b \in B_F$ (resp. $a \in A_F$), $\mathcal{E}$ contains an edge $\overrightarrow{sb}$ (resp. $\overrightarrow{at}$) of weight $0$. From the definition of slack, it follows that the weight of all edges in the augmented residual network in non-negative. The weight of any directed path in the augmented residual network is simply the sum of the weights of its edges.

## 3    EXACT ALGORITHM

We begin by describing an algorithm to compute an optimal transport plan and establish a few of its properties. Our algorithm is based on the primal-dual framework where it constructs a transport plan while maintaining a feasible solution for the dual program for the optimal transport. Then, we show how the intermediate computations in the execution of this algorithm describes the exact OT-profile.

**Initialization Step:** The algorithm picks an initial transport plan $\sigma$ where, for every edge $(a, b) \in A \times B$, $\sigma(a, b) = 0$. Let $\alpha$ (initialized to 0) denote the mass transported by $\sigma$. The algorithm also initializes the dual weights of every vertex $v \in A \cup B$ to 0, i.e., $y(v) = 0$. Note that $\sigma$ and $y(\cdot)$ together form a feasible transport plan.

Our algorithm executes in *phases*. Within each phase there are two *steps*.

**First step (Hungarian Search):** Execute Dijkstra's shortest-path algorithm on the augmented residual network $\mathcal{G}_\sigma$ with $s$ as the source and $t$ as the sink[2]. This algorithm computes the minimum weight path (or shortest path) from $s$ to every vertex in the graph. For any vertex $v \in A \cup B$, let $\ell_v$ denote the weight of the shortest path from $s$ to $t$ in $\mathcal{G}_\sigma$, and thus, $\ell_t$ is the weight of the shortest path from $s$ to $t$. The algorithm then uses these shortest path weights to update the dual weights of vertices. For any vertex $v \in A \cup B$, if $\ell_v \geq \ell_t$, the dual weight of $v$ remains unchanged. Otherwise, if $\ell_v < \ell_t$, we update the dual weight as follows: **(U1):** If $v \in A$, we set $y(v) \leftarrow y(v) - \ell_t + \ell_v$, **(U2):** Otherwise, if $v \in B$, we set $y(v) \leftarrow y(v) + \ell_t - \ell_v$.

**Second step:** Let $\mathcal{A}$ be a graph obtained by adding an additional vertex $s$ to the admissible graph, and, for every free supply vertex $b \in B_F$ adding an edge $(s, b)$ directed from $s$ to $b$. Starting from $s$, we execute a Depth First Search (DFS) in $\mathcal{A}$ to find a path to a free demand vertex. Let $P'$ be a path returned by DFS that starts at $s$ and ends at a free demand vertex $a \in A_F$. Let $P$ be the path that remains after we remove the vertex $s$ from $P'$. Note that $P$ is a directed path in the residual graph starting at a free supply node and ending at a free demand node, i.e., $P$ is an augmenting path. Let $r_P > 0$ be the bottleneck capacity of $P$. We update $\sigma$ by augmenting a mass of $r_P$ along the path $P$. After augmentation, we update $\alpha \leftarrow \alpha + r_P$.

The algorithm iteratively executes phases until $\sigma$ becomes a maximum transport plan (i.e., $\alpha = S$). This completes the description of the algorithm.

The dual updates at the end of the first step of each phase guarantee that the transport plan $\sigma$ along with the updated dual weights remains feasible and there is at least one augmenting path in the admissible graph (See Lemma 7.1 and 7.2 in Appendix for a proof). The second step computes an augmenting path $P$ in the admissible graph and increments the mass transported by augmenting the transport plan along $P$. Using this and the fact that the demands and supplies are rational numbers, we show, in Section 7.1 that the algorithm terminates in finite number of phases with a maximum transport plan. Furthermore, we show that our algorithm maintains (C) as an invariant.

**Lemma 3.1.** *The algorithm terminates after executing a finite number of phases. Furthermore, (C) holds for the duration of the algorithm's execution.*

Before we describe the construction of the OT-profile, we introduce a few notations. Suppose the algorithm described above executes for $q$ phases. Let $\{\sigma_0, \sigma_1, \ldots, \sigma_q\}$ be the partial transport plan maintained by the algorithm where $\sigma_0$ is the initial empty transport plan and $\sigma_i$ is the transport plan at the end of phase $i$. Let $\{\alpha_0 = 0, \alpha_1, \ldots, \alpha_q = S\}$ be such that $\alpha_i$ is the mass transported by $\sigma_i$. For any $v \in A \cup B$, let $y_i(v)$ denote the dual weight of $v$ after phase $i$ and let $y_{\max}^i = \max_{v \in A \cup B} |y_i(v)|$. For $1 \leq i \leq q$, let $P_i$ denote the augmenting path computed in phase $i$ and $r_i$ denote its bottleneck capacity. The following lemma, whose proof is in Section 7.2, will be useful in describing the construction of OT-profile and its first derivative.

**Lemma 3.2.** *For every $1 \leq i \leq q$, we have: (a) $w(\sigma_i) = \sum_{j=1}^{i} r_j y_{\max}^j$, and, (b) $y_{\max}^i = y_{\max}^{i-1} + \ell_t$.*

**Generating the OT-profile $\omega$ and the first derivative $D\omega$:** To generate the OT-profile $\omega$, we set $p_i$ to be a point $(\alpha_i, \sum_{j=1}^{i} r_j y_{\max}^j)$ and set $\omega$ to be the piecewise-linear function given by the sequence of points $\langle p_0, p_1, \ldots, p_q \rangle$ where each consecutive pair in the sequence is connected by a line segment. We set the first derivative function $D\omega$ as follows: For $\alpha = 0$, $D\omega(\alpha) = y_{\max}^1$. For any $\alpha \in (\alpha_{i-1}, \alpha_i]$, we set $D\omega(\alpha) = y_{\max}^i$.

**Correctness of the OT Profile:** Next, we discuss why $\omega$ returned by the algorithm is indeed the OT-profile. Observe that, from Lemma 2.1 and Lemma 3.1, $\sigma_i$ is an $\alpha_i$-optimal partial transport and from Lemma 3.2(a), $\omega(\alpha_i) = w(\sigma_i) = \sum_{j=1}^{i} r_j y_{\max}^j$. Therefore, $p_0, p_1, \ldots, p_q$ are points on the OT-profile $\omega$.

---

[2]Recollect that $s$ and $t$ are special vertices added in the augmented residual network.

**Lemma 3.3.** *For any $0 < i \le q$ and any intermediate value $\alpha \in (\alpha_{i-1}, \alpha_i)$, an $\alpha$-optimal partial transport plan can be obtained by augmenting $\sigma_{i-1}$ by a mass of $(\alpha - \alpha_{i-1})$ along the augmenting path $P_i$. The resulting transport plan has a cost of $\sum_{j=1}^{i-1} r_j y_{\max}^j + (\alpha - \alpha_{i-1}) y_{\max}^i$.*

From Lemma 3.3 (proof in Section 7.3), the point $p_\alpha = (\alpha, \sum_{j-1}^{i-1} r_j y_{\max}^j + (\alpha - \alpha_{i-1}) y_{\max}^i)$ will be on the OT-profile. The set of points $p_\alpha$ as $\alpha$ goes from $\alpha_{i-1}$ to $\alpha_i$ will be the line segment connecting $p_{i-1} = (\alpha_{i-1}, \sum_{j=1}^{i-1} r_j y_{\max}^j)$ and $p_i = (\alpha_i, \sum_{j=1}^{i} r_j y_{\max}^j)$. Note that the slope of this line segment is precisely $y_{\max}^i$. Therefore, we can conclude the following:

**Lemma 3.4.** *For any $\alpha \in (\alpha_{i-1}, \alpha_i]$, $D\omega(\alpha) = y_{\max}^i$.*

From Lemma 3.2(b), $\langle y_{\max}^1, \ldots, y_{\max}^q \rangle$ is a non-decreasing sequence, $\omega$ is a non-decreasing piecewise-linear convex function.

**Efficiency analysis:** Each phase of our algorithm requires executing a single Dijkstra's shortest-path algorithm which takes $O(n^2)$ time. Therefore, the total time taken by the algorithm is $O(qn^2)$. Note that the OT-profile generated by our algorithm also has $q$ points, i.e., $\omega = \langle p_0, p_1, p_2, \ldots, p_q \rangle$. However, if a subsequence $\langle p_i, p_{i+1}, \ldots, p_j \rangle$ are all collinear, one can remove the points that lie between $p_i$ and $p_j$ to obtain a sparser representation of $\omega$. Thus, the number of phases may be higher than the complexity $K$ of $\omega$.

Suppose, for every value of $\alpha \in [0, S]$, we assume that the optimal partial transport is unique. In this case, we show (in the proof of Lemma 3.5 presented in Section 7.4) that $\langle y_{\max}^0, \ldots y_{\max}^q \rangle$ is a strictly increasing sequence and therefore no three points on the OT-profile generated by our algorithm are collinear, implying $q = K$.

**Lemma 3.5.** *Given two distributions $\mu$ and $\nu$ with a support set of $A$ and $B$, suppose the demand (resp. supply) at any point of $A$ (resp. $B$) is a rational number. Also, suppose, for every value of $\alpha \in [0, S]$, the optimal partial transport is unique, then the algorithm described above executes in $O(n^2 K)$ time where $K$ is the complexity of the OT-profile.*

In order to avoid collinear points, one can modify the second step of the algorithm as follows. In the second step of any phase $i$, instead of computing a single augmenting path in the admissible graph, we can compute a maximum flow by using a $\tilde{O}(n^2)$ time algorithm Chen et al. (2022). By doing so, we can guarantee that at the end of the phase $i$, there are no more augmenting paths in the admissible graph. Thus, Step 1 of phase $i + 1$ will adjust the dual weights and create an augmenting path in the admissible graph. Therefore, $\ell_t$ as computed in Step 1 of phase $i + 1$ is positive and from Lemma 3.2(b), we have $y_{\max}^{i+1} > y_{\max}^i$. $y_{\max}^i$ and $y_{\max}^{i+1}$ are slopes of line segments $(p_{i-1}, p_i)$ and $(p_i, p_{i+1})$ respectively. Therefore, we conclude that no three points $p_{i-1}, p_i$, and $p_{i+1}$ are collinear points in the OT-profile and so the number of phase $q$ is exactly equal to the complexity $K$ of the OT-profile.

**Theorem 3.6.** *Given two distributions $\mu$ and $\nu$ with a support set of $A$ and $B$, suppose the demand (resp. supply) at any point of $A$ (resp. $B$) is a rational number. Then, the OT-profile of $\mu$ and $\nu$ can be computed in $\tilde{O}(n^2 K)$ time.*

The proof of the outlier lemma (Lemma 1.1) is presented in the Appendix (See Section 7.5).

# 4   APPROXIMATION ALGORITHM

Next, we show how a single execution of the LMR-algorithm (that has an execution time of $O(n^2/\delta + n/\delta^2)$ can be used to compute a $\delta$-approximate OT-profile with a combinatorial complexity of $O(1/\delta)$. The LMR-algorithm is split into three parts. In the first part, the algorithm will scale-and-round the demands and supplies to integers. In the second part, the algorithm finds an approximate transport plan using a primal-dual method that is similar in style to the exact algorithm described in Section 3. Like in the exact case, the intermediate computations of this part can be used to compute an approximate OT-profile. Finally, in the third part, the transport plan is mapped back to the original demands and supplies.

The second part of the LMR-algorithm maintains a partial transport plan $\sigma$ (initialized to 0) and dual weights $y(\cdot)$ (initialized to 0) that satisfy a relaxed set of feasibility conditions. This part of the

algorithm executes in *phases* and terminates when $\sigma$ becomes a maximum transport plan. Within each phase there are two *steps*. Similar to the exact algorithm, in the first step, the algorithm conducts a Hungarian Search and adjusts the dual weights so that there is at least one augmenting path of admissible edges. In the second step, the algorithm computes many augmenting paths and updates $\sigma$ by augmenting it along all paths computed. At the end of the second step, we guarantee that there is no augmenting path of admissible edges. Lahn et al. (2019) bounded the number of phases to $q = O(1/\delta)$ and the total execution time of the algorithm by $O(n^2/\delta + n/\delta^2)$. Let $\sigma_0$ be the initial transport plan and $\sigma_i$ be the partial transport plan maintained after phase $i$. Let $\alpha_0 = 0$ and $\alpha_i$ be the mass transported by $\sigma_i$ and let $y^i_{\max}$ be the largest dual weight at the end of phase $i$. To approximate the OT-profile, we simply add, for each $0 \leq i \leq q$ the points $p_i = (\alpha_i, w(\sigma_i))$. The approximate OT-profile $\overline{\omega}$ is given by the piece-wise linear function $\langle p_0, p_1, \ldots, p_q \rangle$ where every consecutive pair of points in this sequence is connected by a line segment. Unlike in the exact case, $\overline{\omega}$ is not necessarily convex and its first derivative can be unstable. Nonetheless, we are able to define a step function $D\overline{\omega}$ that satisfies an approximate version of the outlier lemma: $D\overline{\omega}(0) = 0$ and for every $\alpha \in (\alpha_i, \alpha_{i+1}]$, $D\overline{\omega}(\alpha) = y^i_{\max}$. Details of the LMR-algorithm, the approximate version of the OT-profile as well as the approximate outlier lemma are presented in the Appendix (see 8, 8.2.1).

## 5 EXPERIMENTS

**Experimental Setup:** We conduct experiments for outlier detection as well as PU-Learning experiments on synthetic and real world dataset. Our algorithm is implemented in Java and experiments were conducted using Python, and are executed on a machine with 2.1 GHz Intel® Xeon® E5-2683v4 processor with 64 GB of RAM.

### 5.1 OUTLIER DETECTION

In this section, we test the effectiveness of our outlier detection lemma on a real-world dataset. Additional experiments on synthetic datasets are presented in the Appendix 9.4. Our lemma helps identify the inlier mass $\alpha^*$ based on an observable increase in the first derivative function around $\alpha^*$.

**Experiment with Real Data-set:** In this experiment, we follow the setup of Mukherjee et al. (2021). Denote $\mu$ as the set of clean data containing $n$ MNIST digits LeCun (1998) from 0 to 4, and $\nu$ as a mixed set contaminated by digits from 5 to 9. We set $\varepsilon$ as the mixing ratio of outlier data in the whole dataset, i.e., $(1 - \varepsilon)n$ digits in $\nu$ are between 0 and 4 and remaining $\varepsilon n$ fraction are outliers that are between 5 and 9. We utilize the $L_1$ distance (Manhattan distance) as the metric of MNIST digits. We execute two sets of experiments, one with $n = 1k$ and another with $n = 10k$ images. For each set, we consider $\varepsilon = 0.2, 0.25$ and $0.3$. Our algorithm computes the OT-profile $\omega$ and its first derivative $D\omega$. Then, it uses the kneedle method (with a default sensitivity of 1) to catch a sudden rise in the first derivative $D\omega$. Thus, from the outlier lemma, we are able to estimate the value of $\varepsilon$ with $\hat{\varepsilon}$ (we refer to this as the check point). We then label the images as inliers (resp. outliers) if they are matched (resp. free) with respect to the $(1 - \hat{\varepsilon})$-optimal partial transport. See Appendix 9.2 for details regarding the application of the kneedle method. We repeat the experiments 30 times and report the average and standard deviation of the accuracy. We also conduct the same set of experiments for the robust optimal transport algorithm of Mukherjee et al. (2021). The results are presented in Table 1.

Table 1: Outlier Detection Accuracy(left) and Check Point Detected(right) for MNIST

| Method($n$) | $\varepsilon = 0.3$ | $\varepsilon = 0.25$ | $\varepsilon = 0.2$ | Method($n$) | $\varepsilon = 0.3$ | $\varepsilon = 0.25$ | $\varepsilon = 0.2$ |
|---|---|---|---|---|---|---|---|
| ROBOT($1k$) | 0.73 | 0.77 | 0.81 | OT-Profile($1k$) | 0.728 | 0.744 | 0.792 |
| OT-Profile($1k$) | 0.81 | 0.82 | 0.85 | OT-Profile($10k$) | 0.754 | 0.770 | 0.796 |
| ROBOT($10k$) | 0.77 | 0.80 | 0.84 | | | | |
| OT-Profile($10k$) | 0.88 | 0.89 | 0.90 | | | | |

Our outlier detection method consistently achieves a slightly better accuracy (between $4\% - 10\%$). Table 1 (right) shows the check point that is automatically detected by our algorithm. Observe that the check point detected by our algorithm is very close to value of $\varepsilon$ (the actual outlier ratio). Note

that the method of Mukherjee et al. (2021) relies on a hyper-parameter that is obtained by sampling $\mu$. Our method on the other hand, does not rely on any data driven hyperparameters. In the supplemental material, we show a visual sample of the miss-classified MNIST images for ROBOT and OT-profile approaches. We also provide number of errors made on a sample of $n = 2000$ by ROBOT and OT-Profile for each digit.

## 5.2 PU LEARNING

In this section, we apply OT-profile in the context of PU Learning Bekker & Davis (2020). The goal of PU learning is similar to binary classification problem where labelled positive examples (let $n_P$ denote the number of positive examples) make up the training set where as the test set consists of positive and negative examples that are unlabelled (let $n_U$ denote the number of unlabelled examples). In case of multiple classes, similar approach is followed where the target classes are considered as positive ($y = 1$) and all the others are considered as negative ($y = -1$). Following the previous work set up, we applied our approach under the SCAR assumption (selected completely at random) on six datasets from UCI repository, which are: Mushrooms, Shuttle, Pageblocks, USPS, Connect-4 and Spambase and test SAR (selected at random) assumption with colored MNIST. From each dataset, we randomly choose equal number of samples, $n_P = n_U = 800$ for the positive set as well as the unlabelled set. Experiments are run 10 times, and the mean accuracy and running times is reported.

Recently, Chapel et al. (2020) demonstrated that specific optimal partial transport formulations (namely Partial Wasserstein (p-W) and Partial Gromov-Wasserstein (p-GW)) can be used for learning positive unlabelled data. In Chapel et al. (2020), it is assumed that the true proportion of positives in the unlabelled dataset (denoted by $\pi = p(y = 1)$), called *class prior* is known in advance for different datasets. $\pi$ is used to sample the test (unlabelled) set as per the marginal $p(x) = \pi p(x|y = 1) + (1 - \pi)p(x|y = -1)$. Both these methods need the knowledge of $\pi$.

On the other hand, we describe an approach that uses the OT-profile $\omega$ and the first derivative $D\omega$ to solve the PU-learning task without the knowledge of class prior. We refer to this approach as *OTP-wo-prior*. Since the unlabelled data can have samples from multiple classes, there can be several noticable rise in the first derivative $D\omega$. In such a case, we pick the earliest jump to estimate the class prior. Let $\hat{\pi}$ be the class prior estimate returned by our algorithm. We then mark all free points with respect to $\hat{\pi}$-optimal partial transport as negative and all other points are positive. We describe our method to detect a jump in $D\omega$ in the Appendix 9.3. We also describe a method assuming the class prior $\pi$ is known. If $\pi$ is known, we simply mark the free points with respect to the $\pi$-partial transport plan as negative and the remaining points as positive. We refer to this approach as *OTP-w-prior*.

We observed that OTP-wo-prior is both computationally efficient as well as achieves comparable or better accuracy (see Table 2, standard deviation reported in Appendix 8) than the methods p-W and p-GW proposed by Chapel et al. (2020). Moreover, unlike previous methods, our method does not need the knowledge of class prior which arguably is a significant limitation of the work by Chapel et al. (2020). Under SCAR assumption, except for usps and spambase, both OTP-w-prior(italic) and OTP-wo-prior(bold) outperform p-W and p-GW. And detected priors are close to the real value within 0.2% for mushrooms, shuttle, and pageblocks. Under SAR, OTP-w-prior yields the best accuracy, and OTP-wo-prior has similar result with p-W and P-GW.

Table 2: PU learning accuracy rates(left) and execution time(right) in seconds

| dataset | $\pi$ | p-W | p-GW | OTP-w-prior | OTP-wo-prior | $\hat{\pi}$ | p-W | p-GW | OTP-w-prior | OTP-wo-prior |
|---|---|---|---|---|---|---|---|---|---|---|
| mushrooms | 0.518 | 96.1 | 95.2 | *99.8* | **99.7** | 0.517 | 1.2 | 83.9 | 0.6 | 0.5 |
| shuttle | 0.786 | 96.3 | 95.5 | *97.9* | **97.7** | 0.785 | 1.0 | 82.3 | 0.5 | 0.4 |
| pageblocks | 0.898 | 92.4 | 90.6 | *93.1* | **93.1** | 0.925 | 1.4 | 92.1 | 0.3 | 0.3 |
| usps | 0.167 | **98.6** | 95.7 | 97.3 | 92.5 | 0.092 | 2.1 | 80.2 | 8.1 | 7.5 |
| connect-4 | 0.658 | 61.0 | 55.4 | *74.7* | **67.3** | 0.860 | 2.2 | 82.0 | 1.9 | 1.8 |
| spambase | 0.394 | **79.8** | 71.1 | 73.4 | 65.9 | 0.301 | 1.2 | 80.3 | 0.8 | 0.7 |
| mnist | 0.1 | 99.1 | 98.4 | *99.3* | 98.1 | 0.081 | 1.8 | 82.3 | 3.1 | 2.6 |
| colored mnist | 0.1 | 91.6 | 97.5 | *99.3* | **97.6** | 0.076 | 1.9 | 81.8 | 3.2 | 2.9 |

## ACKNOWLEDGEMENT

We would like to acknowledge, Advanced Research Computing (ARC) at Virginia Tech, which provided us with the computational resources used to run the experiments. Research presented in this paper was funded by NSF CCF-1909171 and NSF CCF-2223871. We would like to thank the anonymous reviewers for their useful feedback.

## REPRODUCIBILITY STATEMENT

The code used for the experiments reported in this paper is available at `https://github.com/kaiyiz/Computing-all-optimal-partial-transport`.

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
