# OpenReview forum: "Computing all Optimal Partial Transports"
_ICLR.cc/2023/Conference — ICLR 2023 poster_

### Official Review · Reviewer_Xdgc · 2022-10-17

**Confidence:** 3
**Correctness:** 4
**Technical Novelty And Significance:** 3
**Empirical Novelty And Significance:** 3
**Recommendation:** 8

**Clarity, Quality, Novelty And Reproducibility:**

Full disclosure: I previously reviewed this paper for NeurIPS, where, for various reasons, I was hesitant to recommend acceptance. I feel like the current version is much improved, more streamlined, and easier to follow. I appreciate the authors' efforts, and this is reflected in my recommendation.

**Strength And Weaknesses:**

The proof of the main result goes by a standard dual characterization of OT solutions. The authors then demonstrate a procedure that takes an OT solution that transports total mass $\alpha_1$ from the source measure and produces an OT solution of total mass $\alpha_2$ with $\alpha_1 < \alpha_2$. Moreover, it is shown that for any $\alpha \in (\alpha_1, \alpha_2)$ the OT profile linearly interpolates between the values at $\alpha_1$  and $\alpha_2$. Under a slightly restrictive (but probably necessary) assumption of having all costs rational, the above two properties give a complete description of the OT profile. Relaxing the duality condition affords an approximation algorithm for the OT profile.

Strength:
- The paper studies a relevant problem.
- The introduced algorithms are practical.

Weakness:
- The method is not very novel, even if the results are.

Some comments:
- I would be happy to see a formal definition of combinatorial complexity instead of a footnote. After all, it is a central concept for the algorithms.
- The fact that the involved probabilities need to be rational should be emphasized, maybe in the Theorem's statement.
- In the preliminaries section: "Consider any transport plan $\sigma$ transporting mass from $\nu$ to $\mu$." I think the paper adopts the convention that mass is transported from $\mu$ to $\nu$.
- Section 3, first step: I don't think $\ell_t$ was properly defined.

**Summary Of The Paper:**

The paper studies the optimal transport (OT) profile. That is, the dependence of the OT cost on the amount of mass being transported from the source measure. The main result gives a characterization of the OT profile as a piecewise-linear function. This characterization gives rise to an approximation algorithm that allows to numerically approximate the entire OT profile. The paper also explores a potential application for outlier detection.

**Summary Of The Review:**

I feel that the results are new and interesting. Lately, the Machine Learning community has shown an increasing interest in optimal transport methods, and it is likely that some parts of the community will find the paper interesting and relevant. This is especially true given the more practical nature of some of the results. I am therefore inclined to recommend acceptance.

---

> ### Author Response · Authors · 2022-11-17
> **Response to Reviewer Xdgc**
>
> We thank the reviewer for reviewing the manuscript and providing comments and feedback. Below, we provide responses to the reviewer's queries.
>
> **Comment:** *I would be happy to see a formal definition of combinatorial complexity instead of a footnote. After all, it is a central concept for the algorithms.*
>
> **Our Response:** We have added the definition of combinatorial complexity in both the abstract as well as the main text (See beginning of page 3 under "Our Contributions").
>
> **Comment:** *The fact that the involved probabilities need to be rational should be emphasized, maybe in the Theorem's statement.*
>
> **Our Response:** We have updated Lemma 3.5 and Theorem 3.6 to reflect the fact that demands and supplies are rational numbers.
>
> **Comment:** *In the preliminaries section: "Consider any transport plan $\sigma$ transporting mass from $\nu$ to $\mu$." I think the paper adopts the convention that mass is transported from $\mu$ to $\nu$.*
>
> **Our Response:** Throughout the paper, we use $B$ (the support of $\nu$) as supply points and use $A$ (the support of $\mu$) as demand points. Thus, $\sigma$ transports mass from the supply (support of $\nu$) to the demand (support of $\mu$), i.e., $\nu$ to $\mu$. We have checked the paper carefully and removed any typos related to this convention.
>
> **Comment:** *Section 3, first step: I don't think $\ell_t$ was properly defined.*
>
> **Our Response:** $\ell_t$ is the weight of the shortest path from $s$ to $t$ in the augmented residual network $\mathcal{G}_{\sigma}$. We have added an additional sentence in the algorithm's description that emphasizes the definition of $\ell_t$.

---

### Official Review · Reviewer_HDeV · 2022-10-25

**Confidence:** 3
**Correctness:** 4
**Technical Novelty And Significance:** 3
**Empirical Novelty And Significance:** 3
**Recommendation:** 6

**Clarity, Quality, Novelty And Reproducibility:**

The paper is well-written and clear. The results as a whole look novel and non-trivial.

Minor comments:
- does your algorithm give a $(1\pm\epsilon)$-factor approximation scheme?
- please define what is $n$ in the abstract as well.


**Strength And Weaknesses:**

Strength: the OT problem is non-trivial and has a rich history. The authors give theoretical guarantees of their algorithm before giving experimental evidence of the superiority of their algorithm in detecting outliers.

Weakness: A high-level idea of their theoretical algorithm would have improved the presentation. I think the n^2K bound may be expensive in practice due to a large value of K. The approximation algorithm looks like a straightforward application of Lahn et al.

**Summary Of The Paper:**

In the optimal transport problem, we are given two mass distributions $\mu$ and $\nu$ and the goal is to transport a mass of $\alpha$ from $\nu$ to $\mu$ using a minimal cost with respect to a certain distance such as $\ell_1$. In this paper, the authors look at the problem of computing an optimal OT profile for any particular fixing of $\alpha$ when the underlying distributions are discrete.

The authors first show that the OT profile is a piecewise linear function with a non-decreasing convex function. They first give an exact algorithm of time complexity $n^2K$ where $n$ is the support size of the distributions and $K$ is the number of pieces of the OT profile. Then for any $\delta>0$ they give a $1/\delta$-factor approximate algorithm for the OT profile using $n^2/\delta+n/\delta^2$ time. For designing this approximation algorithm, they use an existing algorithm by Lahn et al.

Then they apply the OT-profile computation to the detection of outliers under a certain reasonable assumption of the outliers. They conduct experiments on the MNIST dataset with outliers and show evidence that their algorithms better the existing algorithm in achieving a better accuracy of detecting outliers.

**Summary Of The Review:**

Due to novelty of the algorithm design and the experimental evidence, I tend to accept the paper.

---

> ### Author Response · Authors · 2022-11-17
> **Response to Reviewer HDeV**
>
> We would like to thank the reviewer for their useful feedback. Please consider our common response as well as an updated manuscript (pages 2 and 3) for an in-depth description of our theoretical contributions.
>
> **Comment:** *A high-level idea of their theoretical algorithm would have improved the presentation.*
>
> **Our Response:** We have added a high-level idea of our algorithm to the paper and included a pseudo-code of our exact algorithm in the Appendix. The algorithm is based on the primal-dual framework where it constructs a transport plan while maintaining a feasible solution to the LP dual of the optimal transport problem. The approximation algorithm also constructs a transport plan while maintaining a feasible solution to a relaxed version of this LP-dual.
>
> **Comment:** *I think the $n^2K$ bound may be expensive in practice due to a large value of $K$.*
>
> **Our Response:** Indeed $K$ can be very large. However, any exact algorithm that computes the entire OT-profile will have to output a function of size $K$. Therefore, a dependence on $K$ is unavoidable. In contrast, our approximation algorithm (LMR-algorithm) has an execution time independent of $K$ and it constructs a compact piece-wise linear approximation of size $\lceil 4/\delta\rceil$ of the OT-profile.
>
> **Comment:** *The approximation algorithm looks like a straightforward application of Lahn et al.*
>
> **Our Response:** Our approximation algorithm is indeed the same as LMR-algorithm. However, we establish several new results about this algorithm.
> * We prove that this algorithm not only produces an approximate OT-cost, but also traces an approximate OT-profile $\overline{\omega}$.
> * The approximate OT-profile $\overline{\omega}$ generated by LMR-algorithm is not a convex function, its first derivative may be very unstable and therefore does not satisfy any approximate version of the Outlier Lemma (Lemma 1.1). Instead, by drawing insights from our exact algorithm, we describe a non-decreasing step function $D\overline{\omega}$ that tracks the evolution of certain dual weights maintained by the LMR-algorithm (see Section 7.2.1). We then show that $D\overline{\omega}$ satisfies an approximate version of the Outlier Lemma (Lemma 7.3). This is also a novel contribution of our work.
>
> **Comment:** *Does your algorithm give a $(1\pm \epsilon)$-factor approximation scheme?*
>
> **Our Response:** Our algorithm does not provide a relative (multiplicative) approximation scheme but only an additive approximation scheme. In fact, similar to the exact algorithm, for small values of $\epsilon$, the combinatorial complexity $K$ of a relative $(1+\epsilon)$ OT-profile can be very large (possibly polynomial in $n$). As a result, any algorithm that computes a relative approximation will have an execution time that depends on $K$.
>
> **Comment:** *Please define what is $n$ in the abstract as well.*
>
> **Our Response:** Thank you for pointing this out. We have now defined $n$ in the abstract.

---

### Official Review · Reviewer_JBGe · 2022-10-26

**Confidence:** 3
**Correctness:** 3
**Technical Novelty And Significance:** 2
**Empirical Novelty And Significance:** 3
**Recommendation:** 6

**Clarity, Quality, Novelty And Reproducibility:**

Apart from some notation issues, the writing is quite clear. As I mentioned, I don't find the work very novel, at least in the theoretical part.

**Strength And Weaknesses:**

Strengths:

-This paper slightly generalizes and appropriately adapts an algorithm for partial optimal transport. This is further used to improve performance in a couple of real-world applications.

Weaknesses:
- The algorithms and techniques seem to be similar and heavily dependent on prior work.
- Regarding prior work, even though the most relevant work is mentioned, the paper is missing a wider overview of prior work on this topic.

Minor comments:
- In lines 1 and 2 of the Introduction I think $A$ and $B$ should be swapped. $A$ should be the support of $\mu$ and $B$ the support of $\nu$
- Page 2, 4th paragraph,1st line: "$\alpha$-approximate"-> "$\delta$-approximate"
- Last paragraph of section 3, line 1: the proof of Lemma 1.1 is not presented there. A reference to the Appendix is missing.

**Summary Of The Paper:**

In this paper, the authors consider the problem of optimal partial transport. In this problem some fraction of a mass distribution over a set of supply nodes, needs to be transported and distributed to a subset of the demand nodes. The amount of mass supplied or demanded by each node as well as the per unit transportation cost between a pair of nodes are given. The goal is to find the optimal way to transport a fraction $\alpha$ of the supply mass to demand nodes.
The authors are interested to find the so called "optimal transport profile" $\omega(\alpha)$ which is the optimal transport as a function of this parameter $\alpha$. The proposed algorithm is a modification of an algorithm by Lahn at' al (LMR algorithm) also using ideas from the Hungarian algorithm like augmenting paths. The modification is needed so that the optimal transport cost if found as a function of $\alpha$ rather than for a fixed value of it. The running time of the algorithm is output sensitive and proportional to the number of linear parts of the output $\omega$, which is shown to be a piecewise-linear function. In particular, the running time is $O(n^2\cdot K)$, where $n$ is the total number of nodes and $K$ the number of linear parts of $\omega$. They also provide a $\delta$-approximation algorithm that outputs an approximate profile $\bar{\omega}$ that has O(1/\delta) linear parts and can be computed in $O(n^2/\delta + n\delta^2)$ time.
The applications mentioned in the paper for these algorithms are Outlier detection and PU learning. The reason why the optimal transport algorithm could be used for the former, is that the existence of outliers would result in a "sudden jump" in the OT-profile $\omega$ since for some sufficiently large value of $\alpha$, some mass has to also be transported through the "outliers".

**Summary Of The Review:**

Although it's an interesting adaptation of prior work, I believe the algorithms and techniques are not novel enough to recommend acceptance.

---

> ### Author Response · Authors · 2022-11-17
> **Response to Reviewer JBGe**
>
> Thank you for the valuable feedback that made us rewrite the introduction of our paper and helped better highlight our contributions. We would like to request you also look at the common response as well as the updated “Our Contributions” part (pages 2 and 3) in the new manuscript.
>
> We also want to point out a misunderstanding in your summary of our paper:
>
>  **Comment:** *The reason why the optimal transport algorithm could be used for the former, is that the existence of outliers would result in a "sudden jump" in the OT-profile $\omega$ since for some sufficiently large value of $\alpha$, some mass has to also be transported through the "outliers".*
>
> **Our Response:** We show that there is a sudden jump in the *first derivative* of the OT-profile. This is different from your claim in your summary that the jump is in the OT-profile itself. In fact, if the outlier mass is small, there may not be a significant jump in OT-profile. Our claim, on the other hand, is that the OT-profile (which is a convex function) sees a big bend when the outliers (irrespective of how much mass the outliers have) start getting transported. We believe that this is a non-trivial result (See proof of Lemma 1.1).
>
> **Comment:** *The algorithms and techniques seem to be similar and heavily dependent on prior work.*
>
> **Our Response:** Please look at our common response where we describe our novel technical contributions.
>
> **Comment:** *Regarding prior work, even though the most relevant work is mentioned, the paper is missing a wider overview of prior work on this topic.*
>
> **Our Response:** We have included some additional references. For instance, we have included references to papers on various techniques for Optimal Transport Approximation Algorithms within the Machine Learning community.
>
> **Comment:** *In lines 1 and 2 of the Introduction I think $A$ and $B$ should be swapped. A should be the support of $\mu$ and B the support of $\nu$.*
>
> **Our Response:** Thank you for pointing this out. We have fixed this typo.
>
> **Comment:** *Page 2, 4th paragraph,1st line: "$\alpha$-approximate" $\rightarrow$ "$\delta$-approximate".*
>
> **Our Response:** We have fixed this typo.
>
> **Comment:** *Last paragraph of section 3, line 1: the proof of Lemma 1.1 is not presented there. A reference to the Appendix is missing.*
>
> **Our Response:** We have added a reference to the appendix.

---

### Author Response · Authors · 2022-11-17
**General Response to All Reviewers**

Thank you for the reviews. We want to take this opportunity to describe the novel theoretical contributions of our work. In the introduction of our earlier submission, we did not fully highlight these contributions. We have updated the manuscript so it correctly reflects these contributions.

From a theoretical standpoint, there are two main contributions:
* We provide the first algorithms for computing (resp. approximating) the OT-profile and its first derivative. Our first novel contribution is to show that a single execution of our exact (resp. approximation) algorithm not only computes the exact (resp. approximate) OT-cost but also traces the entire exact (resp. approximate) OT-profile. While the algorithms are generalizations of known methods, the proof that they trace the entire OT-profile is a new one.
* Currently, there is very limited understanding of OT-profile and its properties. Except for the seminal work by Figalli [1], we are not aware of any significant prior work on analyzing OT-profile. Figalli’s work is for continuous distributions and squared-Euclidean ground distance. We present a new framework to prove mathematical properties of the OT-profile and its first derivative for any ground distance! Using this framework, we identify several properties of the OT-profile including a new outlier lemma (Lemma 1.1).

We now elaborate on the several novel claims within our paper.

* Lemma 2.1 shows that any algorithm that maintains condition (C) as an invariant and incrementally constructs the optimal transport traces the OT-profile. In Lemma 3.1, we show that our exact algorithm maintains (C) as an invariant and therefore, it traces the OT-profile.
* As the OT-profile is traced, certain dual weights maintained by the algorithm capture the first derivative values (Lemma 3.4). By tracking the evolution of these dual weights, we show that the first derivative $D\omega$ is a non-decreasing step function (Lemma 3.2(b) and 3.4) and therefore, $\omega$ is piecewise-linear convex function.
* To analyze the behavior of OT-profile and its first derivative, we track the evolution of these dual weights during the execution of the algorithm. We use Lemma 2.1, 3.2 and 3.4 in a non-trivial way to obtain a proof of the Outlier Lemma (Lemma 1.1).
* In Lemma 7.2, we show that any algorithm that satisfies (C’) as an invariant approximates the OT-profile. We show that the LMR-algorithm satisfies (C’) (Section 7.2.1) and therefore, it traces an approximate OT-profile $\overline{\omega}$.
* However, this approximate OT-profile $\overline{\omega}$ is not a convex function and its first derivative may be very unstable and therefore does not satisfy any approximate version of the Outlier Lemma (Lemma 1.1). Instead, by drawing insights from our exact algorithm, we describe a non-decreasing step function $D\overline{\omega}$ that tracks the evolution of certain dual weights maintained by the LMR-algorithm (see Section 7.2.1). We then show that $D\overline{\omega}$ satisfies an approximate version of the Outlier Lemma (Lemma 7.3).

Our theoretical results in conjunction with the experiments on outlier detection as well as PU-Learning show how OT-profile can be more useful than the OT-cost when it comes to machine learning applications.

References:

[1] Figalli, Alessio. “The optimal partial transport problem." Archive for rational mechanics and analysis 195.2 (2010): 533-560.

---

### Decision · Program_Chairs · 2023-01-20

**Decision:**

Accept: poster

**Justification For Why Not Higher Score:**


 Experiments are carried out on  bit toyish problems (like MNIST or UCI datasets). It would have strongly
supported the paper to have experiments on real-world outlier detection data.


**Justification For Why Not Lower Score:**

The paper proposes a new algorithm for computing all the partial optimal transport. This is a novel and interesting contribution

**Metareview: Summary, Strengths And Weaknesses:**


The paper introduces a new analysis of the optimal transport (OT) profile.
They look at the dependence of the OT cost on the amount of mass being transported from the source measure to the target one.
One of their key results is to show that the OT profile is a piecewise-linear function. Based on this characterization,
they are able to derive an efficient approximation algorithm for computing the entire OT profile.
This algorithm is numerically analyzed and applied for outlier detection.

All reviewers agree that the paper propose a novel and interesting contribution and as such can be accepted for
publication


**Note From Pc:**

if the above contains the word "oral" or "spotlight" please see: "oral" presentation means -> notable-top-5% and "spotlight" means -> notable-top-25%. As stated in our emails, we are disassociating presentation type from AC recommendations